# Influence of the Addition of Extruded Endogenous *Tartary* Buckwheat Starch on Processing and Quality of Gluten-Free Noodles

**DOI:** 10.3390/foods10112693

**Published:** 2021-11-04

**Authors:** Xiao-Miao Han, Jun-Jie Xing, Xiao-Na Guo, Ke-Xue Zhu

**Affiliations:** 1State Key Laboratory of Food Science and Technology, Jiangnan University, 1800 Lihu Avenue, Wuxi 214122, China; 7180112009@stu.jiangnan.edu.cn (X.-M.H.); jjxing@jiangnan.edu.cn (J.-J.X.); xiaonaguo@jiangnan.edu.cn (X.-N.G.); 2School of Food Science and Technology, Jiangnan University, 1800 Lihu Avenue, Wuxi 214122, China

**Keywords:** pasting properties, dough rheology, microstructure, water distribution, gel-entrapped network

## Abstract

Extruded starch could be used as a thickener for food processing due to its pre-gel properties. This study aimed to explore the influence of the addition of extruded endogenous *Tartary* buckwheat starch (ES) on the process and quality of gluten-free noodles. ES was mixed with *Tartary* buckwheat flour in different proportions (10–40%) to prepare the blended flour and noodles. When the content of ES was increased, the swelling power of blended flour at 90 °C had no significant changes, and the decrease in peak viscosity of blended flour was reduced. This indicated that the high-content ES could afford better thermal stability for blended flour and inhibit the swelling ability. The higher level of ES was beneficial to the formation and stabilization of dough, and the improvement of noodle tensile strength. Furthermore, there was no difference in cooking loss between noodles with 30% and 40% ES addition. The microstructure and water distribution of the noodles prepared by blended flour indicated that the gel-entrapped structure organized by the higher content ES could be closely related to the above results. In conclusion, higher ES could contribute to improving the processing properties and quality of noodles.

## 1. Introduction

In recent decades, pseudo-cereals have attracted increasing interest due to the growing realization of their potential as health foods [1]. *Tartary* buckwheat (*Fagopyrum tataricum*) is well-known for its high content of flavonoids such as rutin and isoquercitrin, which exhibit various pharmacological benefits, including antioxidant, anti-obesity, and anti-inflammatory activities [2]. *Tartary* buckwheat is often used as a functional food or food ingredient for health benefits, and it is becoming increasingly popular with consumers. Nevertheless, the absence of gluten in buckwheat flour has limited its development in the food field [3].

The preparation of gluten-free/high-content *Tartary* buckwheat noodles has always attracted the attention of research scholars [4,5]. However, the high content of *Tartary* buckwheat is detrimental to the formation of noodles, which is still a challenge for preparing gluten-free high-content buckwheat noodles [6,7]. Owing to its cold-water-swelling capacity and desirable pasting and texturizing properties, extruded *Tartary* buckwheat starch (ES) could be used as a thickener for gluten-free *Tartary* buckwheat noodles [8,9]. In our previous study, the gluten-free *Tartary* buckwheat noodles were successfully prepared by adding ES (30%), which formed the cold gel [9]. Additionally, in previous pre-experiments, it was found that less ES in the blended flour (ES and TBF) failed to develop gluten-free noodles, while excess ES led to the occurrence of dough sticking during sheeting. Therefore, the amount of ES in gluten-free *Tartary* buckwheat noodles is important for the processing of noodles. The formation of the gel-entrapped network in gluten-free noodles benefits from the cold gelation property of ES, which is closely related to the amount of ES added and was hypothesized to affect the quality of gluten-free noodles. However, studies on the effect of extruded starch addition on the processing and quality of gluten-free noodle sheets have not been investigated. Moreover, the internal relationship among the swelling capacity and viscosity of blended flour, rheological properties of the dough, and the quality of gluten-free noodles needs to be further explored.

In this study, the blended flour obtained by mixing different contents of ES with *Tartary* buckwheat flour was used to prepare the noodles: (i) the solubility, the swelling power, and the pasting characteristics of the blended flour are compared, and (ii) the rheological properties of the dough, the tensile characteristics of fresh and cooked dried noodles, the microstructural changes, and water migration of the noodles are investigated to explore the influence of the addition of extruded endogenous *Tartary* buckwheat starch on the processing and quality of gluten-free noodles.

## 2. Materials and Methods

### 2.1. Materials

*Tartary* buckwheat flour (TBF) (water content: 14.11%, crude protein content 7.93%, ash content 2.21%, starch content 74.00%, free fat content: 2.96%) was purchased from Xichang Qiongchi *Tartary* buckwheat Industry Co., Ltd., Xichang, China.

### 2.2. Preparation of Extruded Starch and Blended Flour

*Tartary* buckwheat starches extracted from buckwheat flour were pre-gelatinized to gain extruded *Tartary* buckwheat starch (ES) by a twin-screw extruder (FMHE 36–24; Hunan Fumaco Food Engineering Technology Co., Ltd., Changsha, China) at 125 °C of core extruded temperature. The extraction method and extrusion parameters specified by Han et al. [9] were used.

The blended flours were prepared by mixing ES with *Tartary* buckwheat flour (TBF) in the ratios of 10:90, 20:80, 30:70, and 40:60 and labeled as 10% E-F, 20% E-F, 30% E-F, and 40% E-F, respectively.

### 2.3. Swelling Power and Solubility Analysis

The swelling power (*g*/*g*) and solubility of TBF, ES, and blended flour were determined according to Anderson, Conway, Pfeifer, and Griffin [10], with slight modification. First, the sample (0.1 g) was mixed with 8 mL of distilled water, kept in a water bath for 30 min at 25 or 90 °C, and then centrifugated at 10,275× *g* for 10 min. The supernatant liquid was poured into a tared evaporating dish and dried at 130 °C for 4 h. The dried supernatant and remaining precipitate were weighed.

### 2.4. Dynamic Viscosity and Pasting Viscosity Analysis

The viscosity of TBF, ES, and blended flour was determined by a rheometer (Discovery HR-3 type rheometer, American TA Instrument Company, New Castle, Delaware, United States). First, the 8% (*w*/*v*) paste was prepared by adding ES, TBF, or blended flour to distilled water and magnetically stirred for five minutes before testing. Second, the paste (1.3 mL) was compressed to 1 mm at a parallel-plate geometry (40 mm) and then measured by varying 0.01–100 s^−1^ of shear rate at 25 °C. The curve fitting power-law equation is as follows:η* = K* λ^n−1^(1)
where η* is the dynamic viscosity, K* is the consistency coefficient, λ is the frequency, and *n** is the power-law index.

Pasting properties of TBF, extruded starch, and blended flour were determined using a Rapid Visco Analyzer (RVA 4500, Perten Instruments, Stockholm, Sweden). The sample (3 g) was added to 25 mL of distilled water in a plastic paddle before the test (corrected on a 14% wet basis). The test method is referenced from Cheng et al. [11].

### 2.5. Dough Rheological Properties

A Mixolab (Mixolab 2; Chopin Technologies, Paris, France) was used to determine the dough rheological properties of *Tartary* buckwheat flour and blended flours. The Mixolab ‘Chopin+’ protocols were chosen for performing dough rheological properties, and all testing parameters were referenced by Moza and Gujral [12].

### 2.6. Preparation of Fresh and Dried Noodles

First, 100 g of blended flour (20% E-F, 30% E-F, and 40% E-F) and 34% distilled water were mixed for 5 min evenly in a pin-type noodle machine. Then, the crumbly dough was put into a plastic zip-lock bag and then rested for 30 min at 30 °C. Second, the crumbly dough was gradually sheeted, and the roller gap was reduced from 2.0 to 0.8 mm with a noodle machine. The noodle sheets were calendared 4 times at 2.0, 1.6, 1.2, and 0.8 mm of roller gaps, respectively. Fresh noodles were obtained by cutting the dough sheet using a noodle cutter.

The fresh noodles were dried according to a predetermined drying procedure. The drying parameters are shown in Appendix A. The dried noodles were obtained after drying the fresh noodles, the moisture of which was 13.5% ± 0.5%.

### 2.7. Cooking Properties of the Dried Noodles

The optimal cooking time, cooking loss, and water absorption dried noodles were measured. Noodles (30 g) were placed in 500 mL of boiling distilled water. The optimal cooking time was defined as the disappearance time of noodles’ white core during the cooking process (sampled every 15 s). The cooking loss and water absorption of noodles were measured according to the method described by Cao et al. [13] and AACC [14]. The formula is as follows:Cooking loss (%) = [M3/(M1 × (1−m))] × 100%(2)
Water absorption rate (%) = [(M2 − M1)/M1] × 100%(3)
where M3 was the weight of the total dry residue lost from the noodles in the cooking water, M1 was the weight of the uncooked dried noodles, m was the water content of the uncooked dried noodles, and M2 was the weight of the cooked noodles.

### 2.8. Tensile Properties of Fresh Noodles and Cooked Noodles

Dried noodles (20 g, 20 cm length) cooked at the optimum cooking time were cooled in cold water for 10 s and then the surface water of the noodles was absorbed by filter paper for the test.

The texture analyzer (TA; XT plus physical property analyzer: Stable Microsystem, London, England) equipped with an A/SPR was used to perform the tensile test of fresh/cooked noodles. The testing parameters were referenced by Han et al. [9]. The tensile force and elasticity distance were gained from the test curve.

### 2.9. Microstructure Characteristics of Fresh Noodles and Cooked Noodles

The fresh or cooked noodles (cooked at the optimum cooking time) were frozen quickly in a −80 °C refrigerator. The frozen noodle was cut into a 40 μm thick section with a cryostat. Then, the sample was stained with 0.25% (m/v) fluorescein isothiocyanate (FITC) acetone solution (488/518 nm) for 1 min. Finally, the dye solution was cleared with deionized water and the samples were observed under the FITC channel (green color) of the instrument (LSM710, Carl Zeiss Microscopy GmbH, Jena, Thuringia, Germany).

### 2.10. NMR Measurement of Fresh Noodles and Cooked Noodles

A low-field nuclear magnetic resonance (LF-NMR) system (MesoMR23-060V-I, Niumag Electronics Technology Co.; Ltd.; Suzhou, China) was used to test water distribution in dough sheets. The fresh (2.5 g) or cooked noodle strings (0.5 g) were cut into small pieces and filled in the 15 mL transparent glass bottle for relaxation testing. The transverse relaxation time, T2, was measured by the Carr–Purcell–Meiboom–Gill (CPMG) pulse sequence. The test parameters of fresh noodles are as follows: TD = 26,992, SW = 100 k Hz, TW = 1500 ms, NS = 4, and NECH = 900. The test parameters of cooked noodles are as follows: TD = 125,030, SW = 200 k Hz, TW = 1500 ms, NS = 4, and NECH = 2000. MultiExpInv analysis software (Niumag Analytical Instrument Corporation, Suzhou, China) was used for CPMG decay curves’ analysis.

### 2.11. Statistical Analysis

Data analysis was conducted by using SPSS 17.0 (SPSS Inc.; Chicago, IL, USA) and Origin 2016 (Origin Lab Corporation, Peoria, IL, USA). A one-way analysis of variance and Tukey’s test were used to establish the significance of differences among the mean values at the 0.05 significance level.

## 3. Results and Discussion

### 3.1. Effect of Extruded Starch on Solubility and Swelling Power of Blended Flour

With the increase of extruded starch content, the swelling of the blended flour (BF) at room temperature increased significantly. Heating promoted starch pasting, resulting in an increased swelling ability of the blended flour, which was significantly higher than that of BF at 25 °C (Figure 1A). The swelling power of extruded starch did not change significantly at 90 and 25 °C, indicating that extruded starch had better thermal stability [15,16]. Furthermore, the swelling power of the blended flour should have increased with the increase in extruded starch content. However, the swelling power of the blended flour did not change significantly with increasing ES. It was speculated that there may be some interaction between ES and TBF that inhibited the thermal swelling power of the blended flour.

The solubility of ES at 25 °C (22.76%) was higher than the solubility of TBF (10.71%) (Figure 1B). The solubility of the blended flour gradually increased with the increase of ES content. Furthermore, the solubility of TBF and 10% E-F and 20% E-F at 90 °C was significantly higher than that of the corresponding blends at 25 °C. Interestingly, the solubility of the 30% E-F and 40% E-F blended flours at 90 °C was not significantly different from the corresponding solubility at 25 °C. These results suggest that higher levels of extruded starch could inhibit the dissolution of soluble substances in the blended flour during heating.

### 3.2. Effect of Extruded Starch on Dynamic Viscosity and Pasting Properties of Blended Flour

The dynamic viscosity curves of the ES, TBF, and blended flour are shown in Figure 2A and the fitting results of viscosity curves are listed in Table 1. The K values of blended flour paste showed a gradual increase, and R^2^ was closer to 1 as the extruded starch increased, which indicated that the consistency of the blended flour increased and exhibited non-Newtonian fluid behavior. In addition, the dynamic viscosity of 10% E-F and 20% E-F pastes showed fluctuations with shear rate, which may be due to the thixotropy of the blended flour pastes [17].

The pasting curves in Figure 2B exhibited that the viscosity of TBF increased rapidly after reaching the pasting temperature. Raw starch granules in an aqueous medium absorb water and swell when heated, then this starch granule network ruptured, increasing its viscosity [15]. Contrastingly, the extruded starch rapidly absorbed water and swelled, resulting in the highest viscosity of extruded starch at the beginning of the RVA test, then remained stabilized during heating. This result was related to the higher thermal stability of the extruded starch, and remained consistent with the results in Figure 1. Furthermore, with the peak/final viscosity of the 10% E-F paste as a reference, the decreases in peak/final viscosity of the other blended flours gradually narrowed as the extruded starch addition increased (Table 1). This indicated that the higher content of ES retarded the reduction of the viscosity of blended flours, which was probably related to the inhibition of starch swelling.

### 3.3. Effect of Extruded Starch on Rheological Properties of Dough

The dough rheological properties of the blended flour were influenced by ES. The water absorption of the dough increased significantly with the increase of ES content (Figure 3A). This result was consistent with the variation pattern of swelling power of the blended flour at room temperature. The dough formation time and stabilization time of the blended flour were significantly reduced after ES addition. This indicated that the addition of ES diluted and disrupted the original dough system of TBF, resulting in a significant decrease in the dough formation time and stabilization time. This may be because the addition of ES increased the water absorption capacity of the dough system, resulting in a reduction in dough formation time [18]. Notably, the dough formation time and stabilization time showed an increasing trend with increasing ES addition, suggesting that higher levels of ES interacted with TBF to inhibit the decrease in dough formation time and stabilization time. In the Appendix A, it was seen that ES could form a homogeneous semi-solid and still maintain a typical gel fracture of honeycomb or sponge-like structure after the freeze-drying [19]. In addition, in our previous study [9], ES formed a gel mass during dough mixing, which could be closely related to the time of dough formation and dough stabilization. Specifically, an increase in ES content led to an increase in the amount of gel in the dough, which could protect the BF dough during shearing, leading to an increase in its dough formation time and stabilization time.

It was reported that protein weakening was associated with the interactions between extruded starch and protein during the mixing process [9,20]. The protein weakening value of the dough should continue to decrease as the ES addition increases (Figure 3B). However, the protein weakening values of the 40% E-F dough were greater than the weakening values of the other three groups of blended flour. This was probably related to the structure of the starch gel formed with TBF. In this structure, the proteins were more accessible to water and more susceptible to temperature damage. The trend of retrogradation of doughs prepared from the blended flour was consistent with those of the proteins’ weakening.

### 3.4. Effect of Extruded Starch on Cooking Characteristics, Tensile Properties, and Microstructure of Noodles

With the increase in the ES from 20% to 30%, the cooking loss of the noodles increased by 7.01%, while the cooking loss of the noodles was not significantly different with the increase in the ES content from 30% to 40%. The cooking time of gluten-free noodles increased significantly, and the water absorption rate of noodles tended to decrease with the increase in the ES content (Figure 4A). The tensile distance and tensile strength of the fresh noodle sheets increased significantly with the addition of ES. In addition, the tensile strength of the cooked noodles also had a similar trend of change (Figure 4B). In a previous study, the formation of a gel-entrapped network of extruded starch in the noodles contributed to enhancing the mechanical strength of the fresh noodles [9]. Therefore, the gel-entrapped structure in gluten-free noodles with higher levels of ES was probably closely related to the above results.

Figure 4C showed that the raw starch granules in 20% E-F noodles were closely packed, and the boundaries of ES were not clear. In 30% E-F noodles, a dense gel-entrapped structure was clearly observed. In particular, the 40% E-F gel boundaries were distinct, and the encapsulation network was wider than that for the 30% E-F sheets. This result was associated with more gel masses wrapping around the raw starch. It was interesting that the gel-entrapped structure was still present in cooked noodles and the distribution of starch remained similar to those of fresh noodles (Figure 4D).

The proportion of ES increased which expanded the opportunity for the starch granules in the noodle sheet to be entrapped by the gel. The ES gel could render the viscoelasticity required for the processing of gluten-free noodles and increase their mechanical tensile strength. It was reported that the cooking loss of noodles was related to the dissolution of insurable substances [20,21]. The gel network structure formed by ES in the noodle sheets was speculated to inhibit the dissolution of insurable substances. In addition, the tightly packed gel structure of the noodles may also inhibit the swelling capacity of the raw starch granules during cooking, thereby reducing the water absorption of the noodles and increasing the cooking time.

### 3.5. Effect of Extruded Starch on Water Migration of Noodles

T_2_ was particularly sensitive to changes in molecular mobility and can be used to characterize diverse water molecules with different mobilities [22]. The T_21_ population consisted of protons with marginal contact with water and was assigned to the signal of starch- and protein-limited exchange protons [23]. T_22_ was closely associated with slightly more mobile protons of different polymers and mainly exchanges with limited water. T_23_ exhibited the highest mobility, and it was composed of free water and weakly bonded OH protons of polymers as well as water [24,25]. T_24_ consisted mainly of protons of bulk water and soluble matter [26,27].

The proton distribution curve suggested that the ES in blended flour mainly affected the weakly bound water and free water of the dough sheet (Figure 5A). The intensity of T_22_ values for 30% E-F and 40% E-F dough sheets was greater than that of the 20% E-F sheets. With the increase in the ES content, T_23_ shifted to a longer time (Table 2). It has been shown that T_22_ and T_23_ are related to exchange protons in the intra-granular (inside the sheet) and extra-granular (outside the sheet) space [27], respectively. Thus, these results suggested that the proton exchange between water and starch and non-starch polysaccharides became frequent. It was probably related to the fact that the ES strengthened interactions between water and starch. With the increase in the ES content, A_21_ and A_23_ of fresh gluten-free noodle sheets exhibited a decreasing trend, while A_22_ exhibited an increasing trend. ES has been reported to convert free water in starch to bound water [28,29,30]. ES could absorb water to form the gel-entrapped structures and promote protein hydration, thereby decreasing the free water content. As a result, A_22_ increased and A_21_ decreased in the 30% E-F and 40% E-F noodles compared to the 20% E-F noodles.

Figure 5B showed that four proton migration peaks could be seen in the noodles after cooking. In particular, the intensities of T_21_, T_23_, and T_24_ were high, indicating that the proton activity in these three regions was more frequent. After the noodles were cooked, the starch microcrystal structure was disturbed, the number of protons in the amorphous region increased [31], and more water migrated into the noodles. Thus, this reflected an increase in the intensities of T_21_ and T_23_ and a decrease in the intensities of T_22_. In addition, T_22_, T_23_, and T_24_ of the cooked noodles tended to migrate to the left and become smaller as the content of extruded starch increased, which indicated that the mobility of free water was becoming smaller. The total amount of free water in the 30% E-F and 40% E-F noodles (A_23_ and A_24_) was lower than that in the 20% E-F. This result suggested that higher levels of ES inhibited the increase of free water content in cooked noodles, which was probably related to the gel-entrapped structure in noodles (Figure 4). Furthermore, 40% E-F cooked noodles had the smallest A_23_ and the largest A_24_. This was most likely related to the increase of water in the soluble matter and was consistent with the trend of cooking loss of noodles.

## 4. Conclusions

Blended flour was prepared by mixing extruded *Tartary* buckwheat starch and *Tartary* buckwheat flour and was used to make gluten-free noodles. The swelling power and solubility of the blended flour at 90 °C were inhibited at higher extruded starch additions. The decrease in peak paste viscosity of the blended flours was retarded with increasing ES content. The gel mass formed by higher ES content in the dough promoted the formation and stabilization of gluten-free dough but enhanced the weakening of proteins. The higher ES content in noodles contributed to higher tensile strength and tensile distance of fresh gluten-free noodles. Microstructural analysis exhibited that the gel-entrapped structure remained in the cooked noodles, which could inhibit the increase of cooking losses and affect the migration of free water in cooked noodles. In conclusion, extruded starches exhibited entrapping ability and thermal stability in the above results, which could contribute to improving the processing properties and quality of noodles. This information is beneficial for expanding the use of extruded *Tartary* buckwheat starch in applications of gluten-free buckwheat products.

## Figures and Tables

**Figure 1 foods-10-02693-f001:**
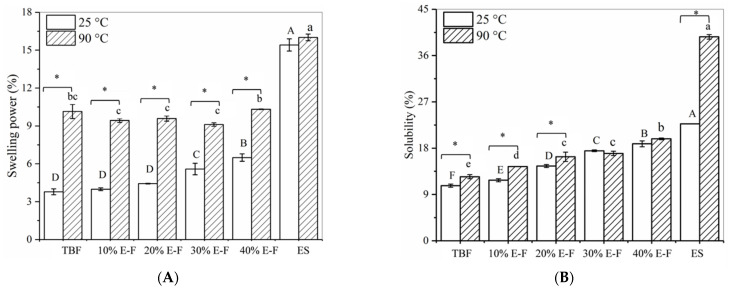
The swelling power (**A**) and the solubility (**B**) of TBF, ES, and blended flour at 25 and 90 °C. ES and TBF represent extruded *Tartary* buckwheat starch and *Tartary* buckwheat flour, respectively. The 10% E-F, 20% E-F, 30% E-F, and 40% E-F represent that the addition of extruded starch in the blended flour was 10%, 20%, 30%, and 40%, respectively. Different letters in the same column represent significant differences (*p* < 0.05). * Represents the significant difference between two adjacent groups (*p* < 0.05).

**Figure 2 foods-10-02693-f002:**
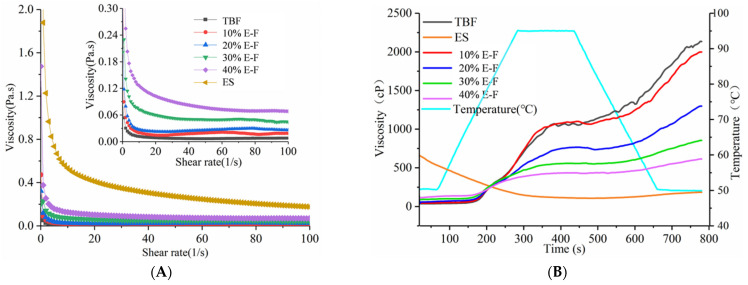
The dynamic (**A**) and pasting viscosity curves (**B**) of TBF, ES, and blended flour. ES and TBF represent extruded *Tartary* buckwheat starch and *Tartary* buckwheat flour, respectively. The 10% E-F, 20% E-F, 30% E-F, and 40% E-F represent that the addition of extruded starch in the blended flour was 10%, 20%, 30%, and 40%, respectively.

**Figure 3 foods-10-02693-f003:**
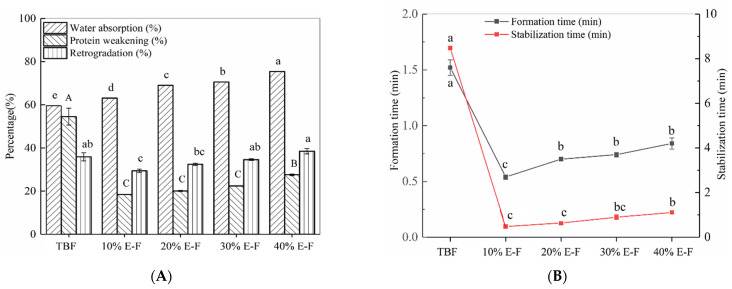
The water absorption rate, protein weakness, and retrogradation rate of dough for *Tartary* buckwheat flour (TBF) and blended flour (**A**). Dough formation time and stabilization time for *Tartary* buckwheat flour (TBF) and blended flour (**B**). The 10% E-F, 20% E-F, 30% E-F, and 40% E-F represent that the addition of extruded starch in the blended flour was 10%, 20%, 30%, and 40%, respectively. Different letters (uppercase and lowercase) in the same group represent significant differences (*p* < 0.05).

**Figure 4 foods-10-02693-f004:**
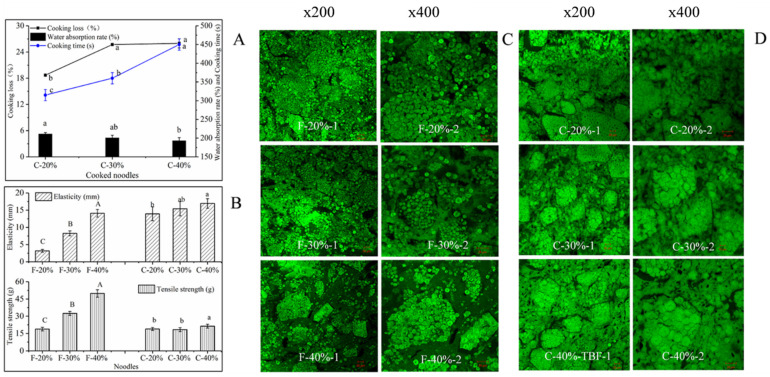
The cooking properties of noodles (**A**), tensile properties (**B**), and microstructure (**C**,**D**) of the fresh and cooked noodles. Different letters (uppercase and lowercase) in the same group represent significant differences (*p* < 0.05). F, C, and TBF represent fresh noodles, cooked noodles, and *Tartary* buckwheat flour, respectively. The 20%, 30%, and 40% represent the addition level of ES in the noodles.

**Figure 5 foods-10-02693-f005:**
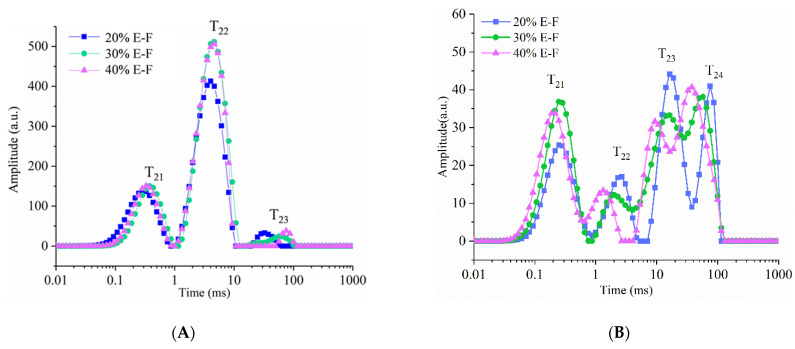
Water distribution curves of fresh noodles (**A**) and cooked noodles (**B**). The 20% E-F, 30% E-F, and 40% E-F represent that the addition of extruded starch in the noodles was 20%, 30%, and 40%, respectively.

**Table 1 foods-10-02693-t001:** The pasting properties of ES, TBF, and blended flours.

Samples	Peak Viscosity (10^−2^ Pa.s)	Tough Viscosity (10^−2^ Pa.s)	Breakdown (10^−2^ Pa.s)	Final Viscosity (10^−2^ Pa.s)	Setback (10^−2^ Pa.s)	K* (10^−2^ Pa.s^n^)
ES	536.00 ± 2.83 ^d^	106.00 ± 1.41 ^e^	430.00 ± 1.41 ^a^	180.50 ± 4.95 ^f^	74.50 ± 3.54 ^f^	155.70 ± 0.76 ^a^
TBF	1072.00 ± 2.83 ^a^	1020.50 ± 2.12 ^a^	51.50 ± 4.95 ^b^	2138.00 ± 5.66 ^a^	1117.50 ± 7.78 ^a^	2.49 ± 0.06 ^e^
10% E-F	1085.00 ± 8.49 ^a^	1028.50 ± 0.71 ^a^	56.50 ± 7.78 ^b^	1996.50 ± 0.71 ^b^	964.50 ± 6.36 ^b^	7.40 ± 0.14 ^d^
20% E-F	755.50 ± 6.36 ^b^	687.50 ± 6.36 ^b^	68.00 ± 0.00 ^b^	1281.00 ± 24.04 ^c^	593.50 ± 17.68 ^c^	7.98 ± 0.15 ^d^
30% E-F	557.50 ± 2.12 ^c^	533.00 ± 12.73 ^c^	24.50 ± 10.61 ^c^	839.50 ± 20.51 ^d^	306.50 ± 7.78 ^d^	14.98 ± 0.15 ^c^
40% E-F	432.00 ± 4.24 ^e^	422.50 ± 3.54 ^d^	9.50 ± 0.71 ^c^	610.50 ± 6.36 ^e^	188.00 ± 2.83 ^e^	30.02 ± 0.17 ^b^

ES and TBF represent the extruded starch and *Tartary* buckwheat flour. The 10% E-F, 20% E-F, 30% E-F, and 40% E-F represent that the addition of extruded starch in the blended flour was 10%, 20%, 30%, and 40%, respectively. Means followed by the different letters in each column were statistically different (*p* < 0.05).

**Table 2 foods-10-02693-t002:** The relaxation times, T2, and peak areas, A2, of gluten-free noodles.

Noodles		T_21_ (ms)	T_22_ (ms)	T_23_ (ms)	T_24_ (ms)	A_21_ (%)	A_22_ (%)	A_23_ (%)	A_24_ (%)
Fresh noodles	F-20% E-F	0.31 ± 0.05 ^a^	4.19 ± 0.26 ^a^	41.26 ± 8.51 ^c^	--	25.78 ± 0.69 ^a^	71.23 ± 0.80 ^b^	3.00 ± 0.19 ^a^	--
	F-30% E-F	0.38 ± 0.04 ^a^	4.82 ± 0.30 ^a^	61.51 ± 4.28 ^ab^	--	21.73 ± 0.91^b^	75.78 ± 0.98 ^a^	2.50 ± 0.25 ^ab^	--
	F-40% E-F	0.38 ± 0.07 ^a^	4.49 ± 0.26 ^a^	76.02 ± 7.50 ^a^	--	21.15 ± 0.38 ^b^	76.68 ± 0.40 ^a^	2.15 ± 0.15 ^b^	--
Cooked noodles	C-20% E-F	0.33 ± 0.07 ^A^	2.92 ± 0.23 ^A^	17.11 ± 2.28 ^A^	61.58 ± 9.22 ^A^	27.57 ± 1.99 ^B^	10.43 ± 2.98 ^A^	33.73 ± 3.65 ^A^	27.97 ± 5.07 ^B^
	C-30% E-F	0.29 ± 0.03 ^A^	2.69 ± 0.49 ^A^	15.77 ± 1.06 ^AB^	58.02 ± 10.62 ^AB^	32.68 ± 1.68 ^AB^	9.13 ± 0.65 ^A^	32.83 ± 2.79 ^A^	25.25 ± 2.56 ^B^
	C-40% E-F	0.25 ± 0.04 ^A^	1.83 ± 0.72 ^A^	9.39 ± 1.31 ^B^	39.53 ± 3.26 ^B^	33.77 ± 2.17 ^A^	6.43 ± 2.14 ^A^	20.63 ± 1.46 ^B^	39.13 ± 0.49 ^A^

Means followed by the different letters in each column are statistically different (*p* < 0.05). The 20% E-F, 30% E-F, and 40% E-F represent that the addition of extruded starch in the noodles was 20%, 30%, and 40%, respectively. F and C represent fresh noodles and cooked noodles, respectively.

## Data Availability

The data that support the findings of this study are available on request from the corresponding author. The data are not publicly available due to privacy or ethical restrictions.

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
