# Peer review of "Influence of the Addition of Extruded Endogenous Tartary Buckwheat Starch on Processing and Quality of Gluten-Free Noodles"

_foods, 2021, doi:10.3390/foods10112693_

Round 1

Reviewer 1 Report

The aim of the papier was to explore the influence of the addition of extruded endogenous Tartary buckwheat starch (in the proportion of 10-40%) on the process and quality of the gluten-free noodles. The research was conducted in a very wide range, i.e. the analyse of the effect of extruded starch addition on the quality of blended flour: solubility and swelling power; dynamic viscosity and pasting properties as well as on the rheological properties od obtained dough and the final effect on cooking characteristics, tensile properties and microstructure of noodle as well as water migration of noodles. The research allowed for the formulation of the final conclusion that extruded starches exhibited entrapping ability and thermal stability in the above results, which could contribute to improving the processing properties and quality of noodles. This information was beneficial for expanding the use of extruded Tartary buckwheat starch in applications of gluten-free buckwheat products.

The manuscript is clear, relevant for the field and presented in a well-structured manner.

I suggest to wirte in italics the a Latin name of Tartary buckweat:  „Fagopyrum tataricum L.”

The cited references are current (mostly 74% within the last 5 years). It includes two self-citations (references: 9 and 13). However references should be improved according to the requirements of the Journal, i.e. bold year of publication in references no: 3, 4, 6, 21, 24; ilatics in the volume number; spasing between characters (i.e. lines: 381, 388, 391, 399, 432).

The manuscript is scientifically sound and the experimental was designed appropriate to test the hypothesis.

The manuscript’s results are reproducible based on the details given in the methods section. However the information about the preparation of the sample to swelling power and solubility analysis should be corrected. Instead of „First, the power (0.1 g) was mixed…” I suggest to write: „First, the sample (0.1 g) was mixed…”.

The figures/tables/images/schemes are appropriate. They properly show the data and are easy to interpret and understand. The data are interpreted appropriately and consistently throughout the manuscript. However, they may be improved:

  • I suggest to change the label of the blended Flour. Instead of „10% E-F, 20% E-F, 30% E-F, and 40% E-F” I suggest to write „10% ES, 20% ES, 30% ES, and 40% ES” in all text, tables and figures captions. Especially that Authors wrote as tables figure captions that: „20%, 30%, and 40% represented the content of adding extruded starch”. The Authors did not explain the abbrevation „E-F” used.
  • Captions under Figure 1, Figure 2 and table 1 do not include the 10% addition of extruded starch
  • The figures should be improved by adding spasing between characters, i.e. Figure 2a, Figure 2b, Figure 3A, Figure 5B.

The conclusions are consistent with the evidence and arguments presented in the manuscript.

The authors declare that there is no conflict of interest regarding the publication of this paper.  Author contributions were explained.

The research was funded by the program for Natural Science Foundation of Jiangsu Province (No BK20190590), the Postdoctoral Research Foundation of China (No 2019 M651706) and the National Natural Science Foundation of China (Grant No. 32072253).

Reviewer 2 Report

Dear Authors,

The demand for gluten-free products is on rising partly due to allergy and partly due to the marketing trend. The current paper " Influence of the addition of extruded endogenous Tartary buckwheat starch on processing and quality of gluten-free noodles" is an attempt in the same line of gluten-free products. However, I have calculated the actual starch which is being added to this product as per the formulations described under section 2.2 para 2 . Taking into consideration the addition of ES and TBF, the actual starch content in the noodles varies from 76.6 % (10:90) to 84.4% (40:60), which us very high. 

1. What was the purpose of adding more starch, when the emphasis should have been to use less starch to make it more healthy?

2. Why starch extraction was done from Buckwheat flour and then added back to it as a concentrated form?

Some other comments :

1. In abstract use "was" instead of "is"....

2. In the introduction also use "was" instead of "is"....

3. line 44, failed to develop instead to prepared...

3. line 44, delete gluten free noodle

4. line 45, delete "roller",.....

5. line 54, delete "gained" use " obtained"....

6. line 54, delete buckwheat...

7. Section 2.2: line 71-72, write "parameters specified by Han et al were used"  instead of " were specifically referred to Han et al" 

8. line 80, delete "power" write "flour" , delete "followed by" write "kept in" 

9. Figure 3(A), use different superscripts for three parameters mentioned in the figure. By using lowercase superscripts in these parameters is confusing to the reader as it shows the comparison between water absorption and protein weakening and retrogradation, which is not the case in actuality. 

10. Figure 3(B), authors have used the common text for both figures e.g different letters in the same column, when there is no column in figure 3(B). Use proper depiction.
